# Teaching English to Linguistically Diverse Students from Migration Backgrounds: From Deficit Perspectives to Pockets of Possibility

Elizabeth J. Erling [1,2,]*, Anouschka Foltz [3], Felicitas Siwik [4] and Michael Brummer [3]

1 Department of Education, University of Vienna, 1010 Vienna, Austria
2 Institute of English and American Studies, University of Vienna, 1090 Vienna, Austria
3 Institute of English Studies, University of Graz, 8010 Graz, Austria; anouschka.foltz@uni-graz.at (A.F.); michael.brummer@outlook.com (M.B.)
4 Department of Multilingualism, Karlsruhe University of Education, 76133 Karlsruhe, Germany; felicitas.siwik@hotmail.de
* Correspondence: elizabeth.erling@univie.ac.at

**Abstract:** This article reports on an interview study with six secondary school LX English teachers working in a part of Austria where there is an above-average number of residents–and thus also students–who are multilingual and come from migration backgrounds. It attempts to extend research on deficit perspectives of multilingual learners from migration backgrounds to the area of LX English learning and to provide insights into a language learning context that is underrepresented in international applied linguistics research, which has tended to focus on elite language learning. The article explores teachers' perceptions of teaching English in this context. We hypothesized that teachers would hold negative beliefs about their students' multilingual backgrounds and practices. The typological analysis of teachers' interview data revealed that teachers did hold some dominant deficit perspectives about their students' multilingualism and language learning; however, it also suggests that teachers are taking on the rudiments of a translanguaging stance that values multilingual practice. The article thus closes by considering how possibility perspectives can be harnessed and extended to foster students' multilingual and multicultural development, with particular regard to LX English language learning.

**Keywords:** multilingualism; LX English language teaching; translanguaging; Austria; LX language learning; social justice

## 1. Introduction

English is a third or additional language for a growing number of students across Europe, including Austria, because of the large numbers of students from diverse linguistic and migration backgrounds in schools. The English language classroom should thus ideally serve as a 'safe space' in which students' multilingualism and multilingual identities are validated and engaged in service of further language learning and the development of intercultural communication (cf. Conteh and Brock 2011). However, deficit perceptions about the multilingualism of students from migration backgrounds have been found amongst teachers in a range of national contexts, even in the context of additional language learning (Haukås 2016; Heyder and Schädlich 2014; Jakisch 2014). In Austria, national reports as well as the media discourses around them frame students' emergent abilities in German as the key contributor to the disparity of outcomes in national and international tests of standards. Commenting on this, the former teacher and journalist Melisa Erkurt argues that negative perceptions of multilingualism in Austria contribute to the lower performance of students from migration backgrounds in school and negatively influence these students' German and heritage language abilities (Erkurt 2020, p. 21). Our research so

far confirms that teachers' beliefs about their students' backgrounds and practices relate to lower levels of achievement in English (Erling et al. 2020, 2021). We have found, for example, that the higher the percentage of multilingual students from migration backgrounds at a school, the more likely it is that teachers believe that their students are not achieving the desired learning outcomes for English. Moreover, despite some developments in teacher education, as in many international contexts (Wernicke et al. 2021), in Austria, there is still an established lack of focus on preparing teachers to work with multilingual learners (Purkarthofer 2016; Cataldo-Schwarzl and Erling 2022).

Because educational discussions about students from migration backgrounds in Austria often place one-sided emphasis on the difficulties of teaching these students, we were surprised to find evidence in a recent study, conducted at small-town schools with a high number of linguistically diverse students from migration backgrounds (Brummer 2019), of a potential counter-narrative to commonly encountered deficit perspectives. In this context, LX English learners[1] seem to be meeting their learning outcomes for English at a higher rate than the national average and we linked that to teachers creating spaces in which students perform their multilingual identities as valued and welcome members of the classroom community. This potential for multilingual students to flourish led us to referring to this educational context as having 'pockets of possibility' (Erling et al. n.d.).

The present article reports on a study that adds insights to the picture emerging: interviews were undertaken in 2018 with six secondary LX English teachers in this small town about their experiences of teaching LX English learners. These teachers were working in three different schools in a part of Austria, near the industrial city of Linz, where there is a distinct history of migration and an above-average number of residents–and thus also students–who are multilingual and come from migration backgrounds. The aim of the research was to explore whether in this linguistically diverse environment teachers were likely to hold deficit perspectives of their students' multilingualism. Secondary schools in this area were chosen in order to extend research on deficit perspectives of multilingual learners from migration backgrounds to the area of LX English learning. Moreover, this language learning context is underrepresented in international applied linguistics research, which has tended to focus on elite language learning (Ortega 2019). We hypothesized that teachers would hold negative beliefs about their students' multilingual backgrounds and practices, that students' other languages would not be seen by teachers as resources for LX English learning, and that this would be reflected in teachers' classroom practices. A typological analysis exploring deficit perspectives and 'pockets of possibility' found that while the teachers do perpetuate some commonly found deficit perspectives of their students' multilingualism and language learning, there is also evidence of the rudiments of a 'translanguaging stance' in which multilingualism is valued and used as a resource for LX English learning (García et al. 2017). While more insights are needed, the findings of this small-scale study add to evidence suggesting the need to promote more asset-based views of multilingualism in education and (language) teacher education. Thus, the article closes by considering how the 'pockets of possibility' identified in teachers' interviews could be harnessed and extended to foster students' multilingual and multicultural development, with particular regard to LX English language learning.

## 2. Literature Review

In this section, we review literature relevant to the discussion of the school achievement of multilingual learners from migration backgrounds in Austria, and teachers' perspectives of students' abilities.

### 2.1. Deficit Perspectives of Multilingualism

Multilingualism is associated with cognitive, social, personal, academic, and professional benefits, with a growing body of research suggesting that a way to raise outcomes of multilingual students is through mobilizing their multilingual repertoires as resources for learning to promote academic success and to boost self-confidence and self-esteem (Cum-

mins 2010; Duarte 2019; García and Kleyn 2016). While research has found that teachers' beliefs about multilingualism in general tend to be positive, as they are aware of research illustrating the benefits of multilingualism, more negative beliefs are often observed when it comes to experiencing multilingual practices in the classroom (De Angelis 2011). This is even the case when considering the value of students' multilingualism for additional language learning (Heyder and Schädlich 2014; Jakisch 2014; Lundberg 2018). Deficit perspectives of multilingualism have been found to persist in many educational systems (Keefer 2017; Mertens 2008). We define deficit perspectives as "restricted and misinformed visions that primarily focus on negative aspects of specific social groups" (Erling et al. n.d.). In the context of students from migration backgrounds, deficit perspectives contribute to beliefs that students' multilingual backgrounds hinder their academic achievement. Such beliefs not only lead to teachers having low expectations of their multilingual students but can also influence their classroom practices (Glock et al. 2019; Pit-ten Cate and Glock 2018, 2019). As a result, teachers may inadvertently convey to their multilingual students the message that they will not be able to meet class expectations (Valencia 2010, p. 9), which in turn can have a profound impact on students' learning, motivation, and academic self-concept, as well as more generally their self-esteem and wellbeing (Borg 2018; Pajares 1992; Tsiplakides and Keramida 2010). Ultimately, such deficit perspectives risk reinforcing educational disparities, lowering multilingual students' educational outcomes, and limiting their future prospects beyond compulsory schooling (MacSwan 2000).

When teachers' beliefs about multilingualism are elicited, their responses suggest widespread misconceptions about the way languages interact in the mind. De Angelis (2011) found that teachers conceive languages as separate entities and seem to believe that one language may somehow interfere with the learning of another, in this case the language of education. When surveying teachers from Austria, Italy, and the UK, she found that over a third of teachers believed that immigrant students must learn one language at a time (35% in Austria), and that a large proportion of teachers believed that the frequent use of the home language delays the learning of the language of education and can be a source of confusion for the immigrant student (35.7% in Austria). Research has also explored the assumption that students' multilingualism often goes hand in hand with a lack of language skills in the language of education (Schmid and Schmidt 2017; Wiese 2011). Such studies have found that students are often framed as being 'doubly semilingual', having only limited ability in all languages in their repertoire. Though the idea of semilingualism has come under much sociolinguistic critique for its narrow conceptualization of literacy and conventional academic language, it continues to have a decisive impact on educational policy and practice (Salö and Karlander 2018). When teachers believe that students are semilingual and have low language ability in all of their languages, this has been found to have a strong negative effect on their expectations for these students, which in turn affects teachers' choices for curricular content and classroom practice (Adair et al. 2017; MacSwan 2000).

In our research, we have found evidence for deficit perspectives of students' multilingualism amongst Austrian English language teachers (Erling et al. 2020, 2021). For example, teachers were more likely to believe that their students are not achieving the standardized learning outcomes if they have a higher percentage of multilingual students in their classrooms. Furthermore, teachers reported little to no use of multilingual pedagogies in the English classroom, even if they teach in linguistically diverse schools. These results suggest that teachers' perspectives are influenced by their students' multilingualism and reflected in their teaching practices.

### 2.2. Perspectives on Multilingualism, Migration, and Education in Austria

Teachers' beliefs about multilingualism are strongly influenced by national discourses about migration and multilingualism (Young 2014). In Austria, one fourth of the population is a first or second-generation immigrant, and in Vienna, 49% of the population comes from a migration background (Statistik Austria 2021). The term "migration background" is an

official statistical category commonly used in Austria (as well as other German-speaking contexts) to refer to people where both parents were born outside Austria (Will 2019). It can refer to both first- or second-generation migrants, and often also implies that these people speak an additional language to German at home. In this article, we use the term 'multilingual students' to refer to linguistically diverse students (who often come from migration backgrounds) who have a range of competences in heritage, national, and foreign languages. Educational issues regarding integrating students from multilingual, migration backgrounds in education have been hotly debated in politics and in the media, increasingly so since 2015 when Europe experienced an influx of migration. Such discourses contribute to problematic stereotypes of multilingual students in urban schools, and particularly in so-called 'Brennpunktschulen', which could be translated as 'schools in disadvantaged areas' (cf. Mohrenberger 2015; Wiesinger 2018).

Educational challenges in Austria are also influenced by the early tracking system, in which students are steered into one of two tracks after only four years of comprehensive schooling. One track is the non-selective middle school (Mittelschule or MS), which comprises grades/years 5 through 8 and aims to equip students for professional training or vocational high school (Handelsakademie or VHS; grades/years 9–12). The other track is the more prestigious, selective high school (Allgemeinbildende Höhere Schule or AHS), which comprises grades 5 through 12 and steers students towards university (BMBWF 2018). This system has come under increasing critique since the rise of the OECD's Programme for International Student Assessment (PISA), which measures 15-year-olds' abilities to use their reading, mathematics, and science knowledge and skills to meet real-life challenges. Findings revealed a surprisingly low performance for Austria, and the early tracking system has been shown to perpetuate disadvantage. Vocational-track schools attract a higher number of students from lower socioeconomic backgrounds as well as from migration and multilingual backgrounds (Herzog-Punzenberger 2017; Schreiner et al. 2020). Students in vocational-track schools perform at lower levels than those in the academic track in all key areas of the curriculum (i.e., German, Mathematics, and English) (Schreiner et al. 2018; Suchań et al. 2019). Moreover, students with a migration background have consistently performed significantly lower than those whose parents were born in Austria.

This has led to multilingual students from multilingual backgrounds often being blamed for the country's underwhelming PISA performance (OECD 2022). However, a differentiated understanding of the findings shows that social status and parents' educational background plays a much more important role in determining performance than migration background or multilingualism (Breit et al. 2016; El-Mafaalani 2021). Despite this, there is a tendency to focus on the role of multilingualism in isolation from the socioeconomic context of students in public discussions of education. This is exemplified in a recent national report on integration (Expertenrat für Integration 2019, p. 33) and the media discourse surrounding it, which has been criticized by the Austrian Association of Applied Linguists for perpetuating deficit perspectives and problem-oriented beliefs about multilingualism (Verbal 2019). In such discussions, the deficit perceptions about multilingualism, semilingualism, and sequential language learning cited above are often perpetuated.

### 2.3. Pockets of Possibility

Translanguaging has arisen as a key theory in contemporary language research to promote the idea that students' multilingualism can be useful in supporting content and language learning and in countering the existing achievement gap between dominant speakers of the language of education and those learning it as an additional language. The implementation of translanguaging pedagogies has therefore been described as potentially transformative in terms of shifting away from deficit perspectives of multilingualism and towards fostering positive multilingual identities and educational equity. García et al. (2017) have argued that in order to support multilingual students' achievement and well-being, teachers need to adopt a translanguaging stance–a staunch belief that students'

whole linguistic repertoires are resources in general and specifically for their learning. This stance also involves creating an overall school and class ecology that is supportive of students' multilingualism and multilingual identities, and supporting them to use their language repertoire to achieve academically (García and Otheguy 2020). Research has shown how enacting a translanguaging stance can lead to the (re)creation of educational spaces that are inclusive of multilingual students' language practices while also supporting their learning of the language of education (Kleyn and García 2019, p. 73). This has been referred to as a 'safe space' (Conteh and Brock 2011) in which educators co-construct meaningful relationships with their learners; support them in performing their identities; and provide space in which students can claim ownership of their own language and everyday knowledge to engage with the curriculum (cf. Brooks 2020). Teachers must believe that translanguaging supports students and their learning, helping them to become more creative and critical (García and Kleyn 2016; García and Wei 2013). When teachers start to change their practice and get a glimmer of such possibilities, they are more likely to ask themselves 'what might be?' (cf. Craft et al. 2012). Such experiences might then lead teachers to further imagine transformation and experiment with their practice. We have termed the rudimentary emergence of such experiences and practices as 'pockets of possibility' (Erling et al. n.d.). These 'pockets' have the potential to propel teachers along a trajectory of developing creative and supportive practices that benefit their students and allow them spaces to construct new knowledge and form positive academic self-concepts as multilinguals.

*2.4. Multilingualism and LX English Language Teaching*

Translanguaging pedagogies entail languages being used flexibly in education, so that students can benefit from content learning across languages as well as improve their competence in all of their languages. Such pedagogies have been shown to be powerful in developing language awareness and metalinguistic awareness and are increasingly recommended for the teaching of linguistically diverse student populations, also in the context of LX English language education (Cenoz and Gorter 2020). The English language classroom is particularly well suited to offer a 'safe space' for LX English learners, especially those otherwise marginalized in the school system. LX English language education can function as a "Wegbereiter" (trailblazer) for the development of multilingual practice and further language learning (Jakisch 2014, p. 202). This can be achieved through activities that cultivate cross-linguistic transfer, metalinguistic awareness, and intercultural competence (Jessner and Allgäuer-Hackl 2020). When students are supported in seeing their multilingual abilities as being beneficial to school achievement, LX English language education can support the development of LX English learners' academic self-concepts and positive multilingual identities.

The fact that almost all students in Austria learn English contributes to the suitability of this subject as a 'safe space': There is a modern foreign language requirement in all school types (vocational and academic) in Austria, with at least 91% of students choosing English as (at least one of) their foreign language requirement(s) (Eurostat 2019). Students at the middle school level are expected to be working at the level of A1–B1 on the Common European Framework of Reference, and high school students are expected to be working at the B1–B2 levels. Students in VHS study English in a professionally related context (e.g., Business English). Most students in AHS learn at least two foreign languages (primarily English and another European language such as French, Italian, or Spanish).

While there is near equal access to English language education in all school types, national standards assessments have shown that there are not equal chances of success in English language learning: AHS students tend to do better in English overall when compared to middle school students (BIFIE 2020, p. 77). Moreover, students who indicated German as their only home language outperformed their peers who indicated having additional home languages by an average of 22–35 points out of a possible 200 to 800 points (although this difference was substantially reduced when socioeconomic class was con-

trolled for, indicating that social class plays an important role in the disparity of outcomes). In the other core subjects of German and mathematics, a larger difference in outcomes between students who indicated that German is their only home language and those with other home languages remains when controlling for social status (Schreiner et al. 2018, p. 52; Suchań et al. 2019, p. 78). Thus, English language education seems to be a core area of the curriculum in which multilingual students from migration backgrounds have the most realistic potential to achieve at the same level of their autochthonous peers, i.e., those from Austria who often come from monolingual German families.

However, in order to level the playing field in English language education, deficit perceptions of LX English learners need to be transformed into possibility perspectives. Moreover, teachers need to be prepared to frame English language learning as a vehicle for further language development and use the English language classroom as a space to celebrate and increase awareness of students' full linguistic repertoires and potential for language learning. While some positive developments have started to occur in this field (Wernicke et al. 2021; Erling and Moore 2021), there is still an established lack of focus on preparing teachers to work with multilingual learners in Austrian teacher education (Purkarthofer 2016; Cataldo-Schwarzl and Erling 2022). A recent review of the English language teacher education curriculum at the University of Vienna, for example, did not uncover a single required course that focused on teaching multilingual learners (Gold 2021). Unsurprisingly, the most recent OECD Teaching and Learning International Survey (TALIS) found that only 31% of teachers in Austria stated that teaching in a multilingual environment was part of their initial teacher education and only 15% felt well prepared to teach multilingual learners (Höller et al. 2019, p. 90). The potential for English language education to support educational equity is thus constrained.

Because educational discussions about students from migration backgrounds in Austria often place one-sided emphasis on the difficulties of teaching them, and because there is a lack of teacher education around multilingualism, our research set out to explore teachers' perceptions of multilingualism and its role in English language education. In this article, we set out to explore the following research questions:

RQ1 What are teachers' perceptions of their students' linguistic diversity?
RQ2 What are their perceptions of multilingual students' abilities and the learning environment?
RQ3 What are their reported classroom practices and perspectives regarding linguistic diversity?

## 3. Methods

In this study, interviews were used to elicit teachers' perceptions, and typological analysis was applied to the data to explore responses. Interviews are a commonly used tool to explore teachers' perceptions of their students and their learning (Cephe and Yalcin 2015). These interviews were undertaken as part of a larger study, which also included questionnaires with students from linguistically diverse, migration backgrounds (see Brummer 2019).

### 3.1. School Contexts

The study was conducted in a small, linguistically diverse town outside the industrial city of Linz, Austria, where linguistic and cultural diversity are the norm in every classroom. The town has about 8,200 residents and is defined by a long history of migration and providing domicile for refugees. The municipality of about 17,500 residents, in which the town is located, has higher than average percentages of residents with migration backgrounds, with 30.6% of residents born outside Austria (compared to 20.1% for Austria as a whole and 16.2% for Upper Austria; Statistik Austria 2020). The vast majority of the municipality's residents with a migration background live in the town where our study was undertaken. Since the end of World War II, the town—now a modern satellite of Linz—has

a history of hosting large numbers of refugees and migrant workers from eastern Europe, Turkey, the former Yugoslavia, Chechnya, and Syria.

Using the last author's (MB) contacts in and knowledge of the area, purposive sampling was adopted to recruit three schools where the research could be conducted. In order to include perspectives from the three main secondary school types in the Austrian system, this study included a middle school (MS) and two high schools: One high school represents a vocational high school focusing on commercial and business education (VHS), the type of high school in which many middle school students continue their education. The second one was an academically oriented high school (AHS). Estimates from the head teachers of these schools revealed that the percentage of students who come from a migration background and speak languages other than German at home was more than 90 percent at the MS, 75 percent at the VHS, and at least 50 percent at the AHS. The MS classifies as a "Brennpunktschule"—a school in a disadvantaged area.

Linguistically diverse schools were chosen as the context for this research for two reasons: First, these schools are often the focus of concern in reports on education. Moreover, there has been a call for applied linguistics to focus more on multilingualism in marginalized and minoritized communities (Ortega 2019). Finally, our findings from a questionnaire study with approximately 200 multilingual students from these three schools (i.e., all students who speak at least one language other than German in the home or with extended family; (Erling et al. n.d.) suggest that these students perform at or above the national average in English, especially at the MS. This suggests that, alongside the expected deficit perspectives, we may find evidence for teachers creating 'pockets of possibility' in this particular context.

### 3.2. Teacher Participants

In order to gain a sense of the range of possible views of teachers in the different secondary school types that exist in Austria, English teachers in the three schools involved in the study were asked to take part in the interviews. Of the 6–10 total English teachers at each school, 6 volunteered to be interviewed: 2 from the MS, 3 from the VHS, and 1 from the AHS (see Table 1). While these participants cannot be considered to be representative of English teachers in these school types, their perspectives provide a glimpse into the experiences of teaching English to linguistically diverse students in this region.

**Table 1.** Overview of the interview participants.

| Teacher | School Type | Sex | Years Teaching at School | Linguistic Repertoire in Addition to English and German |
|---------|-------------|-----|--------------------------|---------------------------------------------------------|
| Felix | MS | male | 1 year | Dutch, Kinyarwanda |
| Jakob | MS | male | 4 years | Italian, French |
| Alena | VHS | female | 6 years | Dutch, Italian, Latin, Czech, Hebrew |
| Beate | VHS | female | 1 year | French, Turkish, Croatian |
| Carla | VHS | female | 3 years | French, Latin, Russian |
| Kerstin | AHS | female | 2 years | Latin, Italian, Spanish |

All names used in the study are pseudonyms. At the time of the interviews, MS teachers Felix and Jakob had been teaching at their school for a year and four years, respectively. In addition to German and English, Jakob is conversational in Italian and French, and Felix is able to communicate in Dutch as well as Kinyarwanda. At the VHS, Alena had been teaching for more than six years; Beate had been teaching for one year; and Carla had been at the school for three years and had worked at an AHS before. Their shared linguistic repertoire included German, English, Dutch, French, Italian, Latin, and some Croatian, Czech, Hebrew, and Russian. The AHS had officially employed Kerstin for

two years, but she also did an internship at that school as part of her pre-service education. In addition to German and English, her linguistic repertoire includes some Latin, Italian, and Spanish. All interviewed teachers grew up monolingually with German and learned their other languages in instructional contexts.

### 3.3. Interviews

In order to explore this study's research questions, interviews were conducted with English teachers by the last author (MB). The goal was to gain insight into teachers' perceptions of their students' abilities in English language learning. The interviews were semi-structured in that the researcher prepared concrete questions, but also allowed the interviewees to put amplified focus on topics they found particularly interesting and which had not been considered by the researcher in advance (Hall 2012, p. 180). The interviews were organized according to themes, including personal information and teachers' linguistic competences. To answer RQ1, questions also explored students' language backgrounds and the number of students who spoke languages other than German at home. To answer RQ2, questions probed multilingual students' performance in the English classroom and classroom dynamics. Questions about English language teaching, designed to answer RQ3, included whether other languages were a resource or a hindrance to learning and whether or not teachers introduced many cultural topics in the classroom (see Appendix A for interview questions). All interviews were conducted in German and recorded on two recording devices to ensure that they were captured.

### 3.4. Ethical Considerations

The researcher who conducted the interviews was originally from the area in which the data were collected and—having personal contacts amongst the students and teachers—took great precautions to ensure his participants' well-being. He used previous contacts to recruit willing participants who were supportive of his research. Written permission to carry out the study was obtained at the outset from the schools' head teachers and the participating teachers. All participants and their schools have been anonymized and pseudonyms have been created to refer to the participants. Based on national laws and university statutes and guidelines, it was not necessary to obtain formal ethics approval. The study, however, adhered to the principles of the Declaration of Helsinki and the British Association for Applied Linguistics Recommendations for Good Practice in Applied Linguistics (BAAL 2021).

### 3.5. Data Preparation and Analysis

The researcher who conducted the interviews also transcribed them, while the other three authors coded the data in MAXQDA 2020 (VERBI Software 2020). In order to uncover patterns and identify themes, they first closely read the transcripts of all the interviews in German and then, as a group, agreed on the main typological categories (as well as subcategories) that would form the analysis. These typologies aligned to a certain extent with the themes that the interview questions sought to explore, although additional categories also emerged. For example, although the intended focus of the interviews was language, teachers also mentioned topics such as students' behavior, the school environment, their own job satisfaction, etc. A code was then created for each typological category and subcategory in MAXQDA. Following the steps recommended for typological analysis (Hatch 2002, p. 153), the first three authors then each coded one interview and checked the coding of another interview in order to enhance reliability. Coding involved assigning relevant statements to the typological categories, where the same statement could be assigned to more than one category. This allowed extracting all statements belonging to one or more categories and facilitates analysis. It was then also considered whether each extract represented a deficit perspective or a possibility perspective. Extracts that support the generalizations being drawn from the data set were selected and are presented below, having been translated into English by the authors.

## 4. Results

In this section, we present the findings from the interviews, exploring the three research questions posed in the study. We first describe students' language repertoires, as perceived by the teachers. We then focus on teachers' descriptions of students' abilities and the school learning environment, including group dynamics in the classroom. Finally, we explore teachers' contributions to the learning environment, including their knowledge of their students' cultural and linguistic backgrounds as an index for valuing students' backgrounds and creating an inclusive learning environment. In each section, we first present the findings from the MS teachers (teachers at the lower secondary level in non-selective vocationally oriented schools); then the results from VHS teachers (the higher-level vocationally oriented secondary school that MS students often go on to); and lastly, the results from the AHS teacher (the academically oriented secondary school that spans the middle and high school years).

### 4.1. Students' Languages

Here, we present the findings about teachers' perceptions of their students' languages by school type, answering RQ1: What are teachers' perceptions of their students' linguistic diversity?

### 4.1.1. Middle School Students' Languages

The MS teachers reported a large number of bi- or multilingual students, with at least 90% of the students speaking another language in addition to German. The prominent countries of heritage were Turkey, the Czech Republic, Bosnia and Herzegovina, Croatia, Serbia, and Slovenia. Syria, Romania, and Hungary were some of the other less common countries of heritage.

### 4.1.2. Vocational High School Students' Languages

The VHS teachers estimated that 75% of their students are bi- or multilingual. This seems particularly apparent among younger students, as Beate had trouble remembering any students who speak German only in the two lower grades. The teachers further reported Romania, Bosnia, and Serbia as the main heritage countries among students with migration backgrounds. Heritage languages most commonly used by the students include Bosnian, Croatian, and Serbian, but also Albanian, Turkish, Dari, and Farsi. VHS teachers reported that students used their heritage languages regularly. Carla described that multilingual students were often conversing in their heritage languages during classroom discussions or to explain tasks to one another. There was some uncertainty, however, regarding students' level of heritage language skills. Teachers were, for example, quite certain that the multilingual students could only speak their heritage languages but were not able to write in them. Moreover, they perceived that their students could not comprehend the grammar of their heritage languages.

### 4.1.3. Academic High School Students' Languages

Kerstin at the AHS reported that the distribution of L1 and LX German speakers could vary depending on the program that they were following. For instance, many of the students enrolled in the scientific program have a migration background. In contrast, the music program has a larger number of autochthonous Austrian students. However, in some of her other classes, the mix of students was fairly even, with roughly 50% of the students coming from diverse backgrounds. She mentioned a particularly positive atmosphere in those classes where there is roughly a 50:50 distribution. She explained that students in these classes gained a lot from each other in terms of cultural understanding and that they were willing, and even interested, to learn what their classmates' heritage languages are like.

*4.2. Students' Abilities and the Learning Environment*

In the following, we explore teachers' perceptions about the school's learning environment and the extent to which they felt that students were performing at their level, also with regard to English language learning, responding to RQ2: What are teachers' perceptions of multilingual students' abilities and the learning environment?

4.2.1. Students' Abilities and the Learning Environment at the Middle School

The MS teachers' opinions about students' performance were mixed. Jakob stated that students' performance is "okay" and "moderate". He related that "There aren't that many that really perform well. So, the level is not so high". While the tendency may be towards lower performance levels, Jakob noted that—while it might not be the majority—there were always students from all backgrounds who were extremely good: "there are also some who are quite high achievers despite the fact that their parents have low levels of education". Jakob commented that as a subject that starts in earnest at the beginning of middle school and that in theory should be less dependent on students' knowledge of German, "English may still be [the subject] where there is nothing to catch up on. Everyone starts the same". Despite the realization that English classes could serve to level the playing field for students with German as an LX, we find evidence of the common deficit perspectives of multilingualism reported on above. Jakob, for example, referred to the common perception that students were semilingual in the following comment: "What you just notice is that when children don't really know their native language, and also don't really speak German, they find it very difficult. If they don't really speak a language, it is very difficult for them". The yard stick applied in this context is clearly literacy rather than spoken language proficiency, as this comment illustrates: "It is also often the case that the parents cannot spell correctly [in their heritage language] and sometimes the children cannot really do it either" (Jakob). Furthermore, there is evidence of the common deficit perspective that adding yet another language overwhelms students: "And yes, then there are a lot of problems with German, and then you add English to the mix" (Jakob).

The MS teachers generally agreed that there is a good atmosphere and collaboration in the classroom, thus a positive learning environment. Jakob stated that "the class community is basically good" and that it "doesn't matter where the parents or grandparents come from, there are always reasonably good class communities". Jakob mentioned that there were a few challenging students, but nothing out of the ordinary. A case of bullying of a new student was discussed but considered to be an exception (Jakob). Felix added: "Given that there are so many nationalities in the classes, the sense of community is really good at this school". Students of different abilities work in the same group, so that "everyone is on one team and sometimes some people who are usually not that good can show that they can do certain things very well. Or they are sometimes pushed a little bit by the others" (Jakob). However, the MS teachers mentioned that there was a tendency for groups to emerge along ethnic lines, especially in the higher grades (Jakob). The teachers believed that these groups emerged because students "stay more and more in their own community" (Felix) outside of the school context. This was considered to be a problem: "And if you had at least a mix in school, and even then, you often see them standing together, for example the students of Turkish origin. Those who then also speak Turkish among themselves, that's a big problem for me, for example. If you mixed it up more, it would be much better" (Felix). To counteract the tendency for groups to emerge along ethnic lines, teachers reported routinely changing the seating arrangement in the class, so that students got to sit with different classmates (Jakob). The teachers also emphasized that students with weaker German skills would benefit if the school had more students with a strong command of German: "it would of course be better if there was more mixing with children who have a better command of German" (Jakob).

The overall positive learning environment at the MS goes along with a high level of teacher satisfaction and a positive school ecology. The teachers described the team as young and dynamic with a recent increase in male teachers (Jakob) and thus an improved

gender balance. Jakob stated that "what is important to me here at this school, and what I have already noticed from other schools where this is not the case, is that the staff work very well together. As a team, so to speak, we're a really good fit. There are no arguments or anything," adding "I'm also very happy with the management. So, I like to be here". Felix agreed: "I couldn't have chosen a better school. On a staff level, too, you are totally supported. You get materials if you have any questions. It makes being a new teacher easier".

4.2.2. Students' Abilities and the Learning Environment at the Vocational High School

At the VHS, perceptions of students' performance were considerably more negative. Beate related that standards at the school were perceived to be low, and that passing grades in English tended to be in the lower range (3 = satisfactory and 4 = sufficient) rather than the upper range (1 = very good and 2 = good). Alena agreed that students' level of performance was low. Another perceived issue was that students at the VHS came to the school (presumably mostly from MS) with already low levels of performance, which lead to a low retention rate: "And, yes, the retention rate, i.e., the students who advance after the first grade [of high school], is not particularly high. It's frighteningly low" (Alena).

Regarding their students' German language skills, the VHS teachers mentioned that their students often had little understanding of German grammar and individual vocabulary items (Carla). This has implications for the English classroom as teachers seemed to rely on students' German knowledge, for example, for vocabulary teaching. For example, Alena mentioned that particularly at the higher levels, it was difficult to teach subtle differences in vocabulary because students did not understand the subtle differences in the German translations of the vocabulary items either. In addition, these three teachers expressed that they could not automatically expect their students to understand the subject matter when it was taught in German (Carla). This relates to an anecdote shared by Beate, who told us that a colleague complained about her own students not being able to follow her lesson due to the high level of German skills she demanded, and that this colleague felt that it was the German teacher's responsibility to fix the problem. Beate further noted, however, that the issue of language proficiency cannot be solved by German teachers alone, as they cannot instantly build up their students' vocabulary. She suggested that teachers of all subjects had to recognize that students for whom German was a second language would not necessarily have the same proficiency as their autochthonous classmates and should respond to this in their lessons.

A particular issue that the VHS teachers mentioned was students' (lack of) motivation. They reported that their students may be capable but that they often did not do their homework, attend classes, or study for exams and that this was what caused the lower levels of performance. Specifically, students were perceived to be satisfied with below average grades (Beate) and as not trying when something was challenging: "They don't even try" (Carla). They reported that a feeling of indifference reigned, what Carla called "ein Wurschtigkeitsgefühl"—the feeling that nothing matters. However, they commented that here, too, were exceptions and that students could become interested in topics and become inspired to engage with various films and/or social media in English outside of school, and that this could help with their English learning (Carla). The VHS teachers expressed some statements about individual cases of negative student behavior, describing their students' behavior as challenging. Despite some negative views, the VHS teachers had generally positive views of group dynamics in the classroom. All three teachers agreed that students in their classes mostly worked well together. Alena emphasized that "in many of my classes there is good cohesion. Many students also support each other". Like Jakob at the MS, Alena viewed this positive atmosphere as "something that characterizes our school".

### 4.2.3. Students' Abilities and the Learning Environment at the Academic High School

At the AHS, students' performance was portrayed more positively. While students reportedly began high school at the AHS with vastly varying levels of English, their knowledge and performance were more homogeneous at the end of high school (which might also be because the students who struggle the most transfer out of the school). In terms of performance, "there are surprises in both directions," Kerstin reported, regardless of students' heritage languages. As at the VHS, problems were noted in terms of literacy, specifically spelling, where Kerstin noted that LX-German students, but not L1-German students, had a tendency to capitalize English common nouns and suggests that this was "probably also due to confusion because in German we capitalize nouns [ . . . ] and I can imagine that there will then be confusion". In the AHS interview, there was no mention of difficult student behavior. Similar to the VHS teachers, views of group dynamics were generally positive. Students were reported to be "ready and interested in what other people's native language is like" and classes were "really completely mixed". Kerstin further emphasized that students speaking their heritage languages amongst themselves was not an issue because "in any case, only English is spoken in English classes".

### 4.3. *Teachers' Practices and Perspectives Regarding Linguistic Diversity*

This section explores teachers' reported classroom practices and their engagement with students' languages and cultural topics in the classroom, providing insights into RQ3: What are teachers' reported classroom practices and perspectives regarding linguistic diversity?

### 4.3.1. Middle School Teachers' Practices and Perspectives

As mentioned earlier, MS teachers generally had good knowledge of their students' cultural and linguistic backgrounds and could list the countries from which their multilingual students come. However, they did not provide information as to whether they made use of this information in the classroom. There were no examples of teachers' recounting that they used students' languages—or allowed students to use them—in the classroom to support learning. The teachers reported that they only taught cultural topics when they came up in the textbooks—but not extensively—and that they mostly focused on topics from English-speaking countries, with the examples of England, Ireland, and Australia given.

Knowledge about students' cultural and linguistic backgrounds varied among the VHS teachers. Beate and Carla knew "the exact origin of only a few [students]". Carla added: "I mean, I didn't ask them directly either. Well, that hasn't been part of class, [discussing] where exactly they come from". Alena, in contrast, was very interested in her students' backgrounds and knew where many of them come from as well as which languages they speak. Her interest seemed to be strengthened by the fact that she had visited Albania and Kosovo—a number of students' heritage countries—and had seen for herself the type of places that many of them might come from. She recounted that having been there and being familiar with the context helped her to remember students' names and where they were from. Alena added "It's also nice to have a conversation with the students: 'Well, where are you from exactly [ . . . ]'?".

### 4.3.2. Vocational High School Teachers' Practices and Perspectives

Although there was some interest in students' migration countries, VHS teachers did not report having competence in students' heritage languages (even though some basic abilities in Czech and Russian were mentioned). Beate stated that since so many students come from Turkey and the former Yugoslavia, she thought she would have picked up a few phrases in their languages; however, although she hears these languages regularly, she has not picked up anything. She related that she knows the oft-used phrase "hajde", which means something akin to "let's go," appears in German rap songs, and is the title of a popular song, but she could not cite any other examples. Alena noted that her experience

was that there was no point in using students' heritage languages to enhance learning as they themselves had no understanding of these languages: "they have very, very little understanding of German grammar and then absolutely no understanding of their own mother tongues". Despite this, all three VHS teachers saw a clear advantage in their students' multilingual abilities and felt that it was something that should be emphasized more. Beate believed that—as populations become increasingly diverse—the demand for people who speak multiple languages increases and the many linguistically diverse students at their school should be able to take advantage of this. Alena added that these qualities were not adequately conveyed to multilingual students in schools or the administrative bodies, where their linguistic repertoire could be of value.

With regard to the inclusion of cultural topics, Beate reported that she regularly builds in cultural topics because she finds that students are really interested in them, and the cultural contexts that she mentioned are England and the United States. Alena, however, said that this was something she does not tend to do as she finds that students have no connection to other countries' cultures because they have never been to England or the US. Moreover, she argued that because many students will not use English in the US or England, but use it more as a lingua franca, she finds it inappropriate to focus too much on cultural topics. Carla agreed with Alena, saying that treating cultural topics such as Christmas at their school can be difficult (perhaps because of students' diverse religious backgrounds), but also because there is often not time to explore such topics.

### 4.3.3. Academic High School Teachers' Practices and Perspectives

Kerstin at the AHS seemed to have excellent knowledge of her students' cultural backgrounds and even integrated her students' cultures and languages into classroom practice. For example, students in one of her classes did presentations on celebrities and many students chose a celebrity from their heritage culture:

> "And then you can tell that a lot of students then choose personalities who have something to do with their culture, be it musicians, Serbian musicians, for example, we had a lot [ . . . ]. And then you noticed that the others were listening attentively and that they were really interested".

Kerstin also remembered having ten minutes of class time left with nothing to do:

> "And that's when I got the idea that we could all take one sentence, 'Hello, how are you?' and then write this sentence on the board in the respective home languages. And that worked really well. They remembered that [sentence] and they still know it today. A year later, they can still do the sentences and they were totally fascinated by what the home languages of the other classmates look like".

Kerstin emphasized the importance of being open to other cultures and languages: "the greatest chances are really that one is open to other cultures and other perspectives, to other approaches, and I personally find that to be really valuable that students become more open to other cultures. That they learn to understand that there are people who approach things differently than they would".

## 5. Discussion

Having presented the results of the interviews in terms of the three research questions, we now consider the extent to which the teacher interviews from these three different school types provide evidence of deficit perspectives and pockets of possibility in LX English language education.

### 5.1. Deficit Perspectives

As could be predicted by the literature review informing this study, one of the most dominant deficit perspectives found in teachers' interviews was the idea that students were semilingual in German and their heritage languages, and that this had negative consequences for additional language learning. In this context, teachers' beliefs about

language knowledge seem to revolve around literacy. Literacy is seen as a hallmark of "knowing" a language, such that parents and students who are perceived as not being literate in their heritage languages are considered to not properly know these languages. This is interesting because, first, the vast majority of parents are most likely L1 speakers of the heritage languages, and as such highly proficient speakers—and in many cases literate in these languages. Presumably a sizable portion of LX German speakers are also proficient speakers of their heritage languages, and the fact that Kerstin could elicit a sentence in her students' home languages and–presumably with their help–write these sentences on the board suggests that at least some of the LX German students have at least emergent literacy in their heritage languages. However, many of them may be speakers of non-standard varieties or—given their migration context—commonly codeswitch or translanguage (cf. MacSwan 2000). There are also students in middle schools with refugee backgrounds, who indeed may have missed out on years of formal education and thus who are still developing formal literacy. These are, of course, special cases. However, despite not having any detailed knowledge of students' proficiencies and repertoires, or linguistic knowledge of students' heritage languages, teachers make judgments about their students' knowledge of grammar of their heritage languages. Observed difficulties with the grammar of German and/or English seem to be transferred into perceived difficulties with the grammar of students' heritage languages, and grammar seems to be conceptualized as belonging to written rather than spoken language. Second, literacy in its most basic sense is typically acquired in school in the Austrian context, not in the home, such that autochthonous students typically learn to read and write German at school. Despite this, teachers seem to hold implicit assumptions that students who are LX German speakers should be literate in their heritage languages and that, presumably, parents–not schools–should be the ones to ensure home language literacy. This focus on literacy thus at best underestimates and at worst misrepresents students' home language skills and puts additional pressures on parents of LX German speakers, making them responsible for students' perceived underachievement.

As could also be expected from the literature review, we identified in the interviews beliefs of LX English as an additional burden for students with emergent German (e.g., when Jakob notes "there are a lot of problems with German, and then you add English to the mix"). These beliefs are informed by commonly held notions that languages should be learned separately and sequentially, such that adding an additional language is seen as problematic if students are perceived not to have sufficiently mastered their other languages. Such beliefs echo outdated perspectives that growing up with more than one language overwhelms children or that learning more than one language simultaneously confuses children (see Genesee 2015; Guiberson 2013) for arguments dispelling these myths). However, such beliefs are in line with previous studies in an Austrian context, discussed above, where De Angelis (2011, p. 222) found that 35% of Austrian teachers agreed or strongly agreed with the statement that immigrant students "must learn one language at a time."

A further prominent deficit perspective held by teachers in all school types is that students' heritage languages are a problem if they are spoken by too many classroom participants. Teachers are of the opinion that it is better not to have large groups of students who come from the same language background. This is despite the fact that teachers observe students with the same heritage language background helping each other using the heritage language. Thus, students spontaneously use their home languages as a resource when they can, but this potential remains untapped. Indeed, there was no mention in the interviews of teachers' allowing their students to use their heritage languages as a resource for English language learning. The teachers also seemed to be concerned that students tend to stay in friendship groups that are based on ethnic and linguistic lines, and hence do not develop their German abilities or have opportunities to integrate with students from other backgrounds. They seemed hesitant to allow the use of other languages at school, as they feared that this will lead to the formation of cliques and the separation of certain language groups, as found elsewhere (cf. Van Der Wildt et al. 2015). The low numbers

of autochthonous Austrian students at these schools means that many students have limited opportunities to develop friendships with autochthonous students–an issue that has been noted about school systems across Europe (Eurydice 2019). The teachers noted, however, that–for the most part–students successfully navigate relationships across ethnic, linguistic, and class boundaries. Policies that allow for more mixed schools and classrooms might better allow learners from different socio-economic and ethnic backgrounds to learn together and from each other, thus improving language learning and educational outcomes more broadly, as well as fostering intercultural communication, inclusion, and social cohesion (cf. European Commission 2020). The dominance of German in the Austrian school system cannot be contested, and research has shown that students from migration backgrounds are keenly aware of the high status of the language of education and of the need and benefit of learning this language–and indeed they tend to be motivated to do so (Alisaari et al. 2019; Gogolin 1994). However, research has established that welcoming multilingualism in school does not tend to encourage segregation of different groups. In fact, same-language friendship groups are less likely to dominate in school contexts where there are tolerant practices towards multilingualism (Van Der Wildt et al. 2015, p. 180). Tolerance seems to encourage multilingual students to show positive interest in each other and each other's languages and could thus be a first step towards encouraging interlinguistic friendships—as experienced by Kerstin in her teaching context, one of the pockets of possibility discussed further below.

### 5.2. Pockets of Possibilities

At these schools, the teachers reported that there is a friendly and open class atmosphere, which indicates a learning environment full of possibility. In line with this, teachers expressed a general interest in students' linguistic and cultural backgrounds and showed detailed knowledge of the rich language diversity at their school. This welcoming environment and positive rapport are also beneficial to creating a safe and motivating atmosphere–which has been found to be essential for positive language learning (Oxford 2016).

As was also found in our related research (Erling et al. 2020, 2021), teachers in this study did not report using multilingual practice in English language teaching and tended not to use students' linguistic repertoires as a resource for English language learning. An exception to this was Kerstin, the AHS teacher who reported the experience of integrating students' heritage cultures into classroom activities with great enthusiasm. Assignments which allowed students to talk about celebrities from their heritage cultures (in English) and teach each other words and phrases from their heritage languages seemed to motivate the students beyond the teacher's expectations, presumably because they allowed students to share their funds of identity (Esteban-Guitart 2016) and to celebrate–instead of hide–their language repertoires. She did not mention what prompted her to experiment with these activities–if it was something she had learned through formal teacher education or had spontaneously thought of on her own. However, Kerstin's enthusiasm about these activities suggests that she will continue to create activities that allow students to draw on their out-of-school knowledge and share with their peers, and to develop her practice along these lines. Offering more formal teacher education opportunities in this direction would only support this.

The teachers at the MS and VHS, however, tended to avoid cultural activities out of fear of overwhelming weaker students. Moreover, these teachers position cultural activities as strongly based on English-speaking countries only, which they felt were either not appropriate for their learners (who were unlikely to have visited those countries) or there was not sufficient time to cover them due to the demands of the curriculum. This is in line with work from other contexts suggesting that students who are already perceived as poorer performers are not provided certain learning experiences, in our case the active integration of cultural work. This is often because teachers believe that underachieving students cannot handle "sophisticated learning experiences" (Adair et al. 2017, p. 309). Elsewhere, we found that teachers' engagement with cultural topics positively affected students' grades,

especially at the middle school level (Erling et al. n.d.). Thus, cultural topics may provide unexplored entries into intercultural comparisons and students' linguistic and cultural diversity.

Teachers' awareness of their students' needs for English as a lingua franca is–in fact–in line with research that shows that students are far more likely to use English with other non-native speakers than they are with native speakers (Seidlhofer 2011) and is thus positive. However, that English is a global lingua franca and that "inner circle countries" should no longer dominate the curriculum does not necessarily mean that topics about English-speaking cultures have no place in the classroom, or that cultural content should be excluded altogether. Indeed, Baker (2012, 2015) has provided research-informed insights into how intercultural awareness can be promoted in English language teaching and guidelines for activities that investigate the relationships between culture, language, and communication in the classroom. These approaches offer alternatives to essentialist national representations of culture and include exploring the complexity of local cultures and the cultural diversity in English-speaking countries, as well as challenging cultural representations in the media or in language learning materials. Such activities allow students the opportunity to draw on their cultural knowledge and to develop their criticality and intercultural competence, a goal of the Austrian English language curriculum for all school types.

Our work has established that students' feeling comfortable in the classroom has a positive impact on their LX English learning (Erling et al. n.d.), and we have argued that this feeling of comfort can be attributed to the positive school environment found at these linguistically diverse schools. This finding seems to be confirmed by the teacher interviews: Even though the teachers do not engage in multilingual practice, they all are generally aware of their students' language and cultural backgrounds. This seems to be particularly important for students at the middle school level, where younger students may need support in developing a positive academic self-concept and multilingual identity, particularly because of their awareness of having been tracked into a less academically oriented form of schooling. Indeed, it seems to be the case in our current study that the teachers at the MS were most aware of their students' cultural and linguistic backgrounds, and this finding–along with the positive school climate and strong teamwork and leadership–seems to be contributing to the comparative success of the school. While the VHS teachers were less knowledgeable of students' backgrounds, Alena recounted the positive difference that visiting the country from which some of her students come had made. Providing teachers with more resources to connect with their students from different backgrounds and allowing students to showcase their out-of-school knowledge and experience might further enhance well-being, motivation, and performance at such schools. While there is little in the formal Austrian teacher education system which supports the development of this stance, this interview set suggests that some teachers have developed it organically. This may be because linguistic and cultural diversity has been integral to the history and development of the town and teachers at such schools have no other option but to embrace an ecology that is supportive of multilingualism.

The most positive perspectives on multilingual students come from the teacher at the AHS. Such positive beliefs may not be surprising given that the school caters to students who were already performing at higher levels at the end of elementary school and were thus tracked into this educational strand. Moreover, there is a lower percentage of multilingual students from migration backgrounds at this school, which has been found to influence teachers' beliefs about their multilingual students (Erling et al. 2020).

The pockets of possibility uncovered through this study suggest that the translanguaging stance emerging can and should be captured and enhanced—both with regard to educational practice generally and English language learning specifically (cf. Erling and Moore 2021; García and Kleyn 2016). However, given that deficit perspectives are present, even amongst these teachers who are fully committed to the education of multilingual students from migration backgrounds, this study confirms the need identified in the literature

review for teacher education in Austria to equip teachers with research-informed notions of multilingualism and multilingual practice, and to encourage teachers to reflect on their "common sense assumptions," which shape their understanding of students' practices and abilities (cf. Cataldo–Schwarzl and Erling 2022). The study suggests a need to promote more asset-based views of multilingualism in education and (language) teacher education, in which the focus lies on valuing the linguistic resources students bring with them to school and using the broad range of diversity inside the classroom as an advantage for further (language) learning. Pre- and in-service teacher education initiatives should support educators in critically examining their own deficit perspectives and practices and draw attention to the continued imperative of disrupting them (cf. Keddie 2011).

## 6. Conclusions

Given the deficit perspectives of multilingualism commonly found in Austrian and European schools more generally, we hypothesized at the outset of this study that teachers would hold negative beliefs about their students' multilingual backgrounds and practices, that students' other languages would not be seen by teachers as resources for English learning, and that this would be reflected in teachers' classroom practices. While this hypothesis was confirmed to a certain extent, we also found positive attitudes and pockets of possibility for good practice in teaching English with multilingual students.

Deficit perspectives that persisted in teachers' interviews included the idea that multilingual students from migration backgrounds were largely semilingual, that they had limited literacy skills in both German and their heritage languages, and that this presented them with difficulties in terms of adding English to their repertoires. Pockets of possibility included that many teachers appeared to be highly aware of students' linguistic and cultural background–and this awareness amongst teachers (even in absence of knowledge of those languages) seems to be powerful in terms of students' sense of well-being in the classroom. Teachers were also being inspired to travel to students' heritage countries on their vacation and to bring this knowledge back to the classroom to better connect with students. There was an example of a teacher trialing activities in which students drew on their out-of-school cultural and linguistic knowledge to share with their peers. She noted how enthusiastically students responded to opportunities such as this. Such experiences will hopefully spawn further "what if?" possibility thinking, as promoted by Craft et al. (2012), which will allow teachers to imagine further embracing multilingual pedagogies.

This study provides insight into a language learning context that has been underrepresented in international applied linguistics research (i.e., a non-selective English language teaching context in a small, linguistically diverse Austrian town). Further research into the experiences of English teachers in linguistically diverse schools is clearly needed. However, the results of this small-scale study suggest that if teachers are provided with resources to share and further develop such practices through pre- and in-service education, this could lead to positive changes in English language education. Embracing a translanguaging stance may allow English to get closer to achieving its potential as a curriculum subject where there is a level playing field for all students. Schools such as the MS featured in this study, where multilingualism and diversity are taken as the norm and students are performing above the national average in English, might be celebrated. They could thus serve as catalysts for further good practice and for embracing multilingual pedagogies. While these schools operate within a stratified education system shaped by Austria's wider socioeconomic and political contexts (Herzog-Punzenberger 2017), they offer some potential for developing good practices in English language education. Further exploration of such sites of English language learning could provide a better understanding of what–beyond teachers' beliefs–facilitates (and shuts down) the creation of 'pockets of possibility'. Such pockets could then be harnessed and extended to develop pedagogical practices to foster students' multilingual and multicultural development, with particular regard to English language learning.

**Author Contributions:** Conceptualization, E.J.E. and M.B.; methodology, E.J.E., A.F., F.S. and M.B.; formal analysis, E.J.E., A.F. and F.S.; investigation, M.B.; data curation, E.J.E., A.F. and F.S.; writing—original draft preparation, E.J.E. and A.F.; writing—review and editing, E.J.E., A.F., F.S. and M.B.; supervision, E.J.E.; project administration, E.J.E. and M.B. All authors have read and agreed to the published version of the manuscript.

**Funding:** This research received no external funding.

**Institutional Review Board Statement:** Based on national laws and university statutes and guidelines, it was not necessary to obtain formal ethics approval. The study, however, adhered to the principles of the Declaration of Helsinki and the British Association for Applied Linguistics Recommendations for Good Practice in Applied Linguistics (BAAL 2021).

**Informed Consent Statement:** Informed consent was obtained from all subjects involved in the study.

**Data Availability Statement:** The data presented in this study are available upon reasonable request from the corresponding author. The data are not publicly available because they contain potentially identifying information.

**Acknowledgments:** The authors would like to thank all of the participants for sharing their views with them. The authors would also like the thank the University of Vienna for providing the Open Access Funding for this article.

**Conflicts of Interest:** The authors declare no conflict of interest.

## Appendix A

English translation of the interview questions for LX English teachers. Information in parentheses states which sets of questions relate to which research question.

### Subjects and experience in school (teacher demographics)

1. What are your other subjects?
2. Do you have a preference for one of your teaching subjects? Why?
3. How long have you been teaching at this school? Have you previously taught at another school?
4. In principle, how do you like teaching at this school?

### Languages (teacher demographics)

1. How many and which languages can you speak? At what level?
2. From which languages do you only know single words or phrases? Can you give some examples?

### School, students, languages (RQ1)

1. How many students do you teach per class on average? How many of them are bilingual or multilingual?
2. In which years do you find the largest number of bilingual students? In which do you find the smallest number?
3. Do you know the cultural background of your bilingual or multilingual students?
4. To which language families do the languages of your bilingual students belong? (Romance, Slavic, Baltic, Turkic, Arabic, . . . )

### Changes (RQ1)

1. Has the number of bilingual students in your classes changed in the last few years? If so, to what extent?
2. Were there any other changes?

### Performance (RQ2)

1. How would you evaluate the performance of your students in those classes?
2. Are the achievements in the current classes different from a few years ago? Do you see a performance increase, decrease, or stability?
3. In your opinion, what could be the causes of changes or stability in performance?

*Classroom dynamics (RQ2)*

1.  How do you evaluate the community and dynamics in your classes? Are there only individual groups of friends on average or is there a comprehensive class cohesion?
2.  In the case of individual groups: Can you see patterns here, according to which principles these groups are created? Developed? (performance, cultural background, . . . )
3.  To what extent do the class dynamics influence your organization and implementation of partner or group work?

*English language learning (RQ3)*

1.  To what extent do you incorporate cultural issues into your English classes? Do you also incorporate the cultural knowledge/experiences of bilingual students when it comes to cultural topics? Why (not)?
2.  Are there any languages that you believe will make learning the English language easier? Why? Experience?
3.  Are there any languages that you believe make English difficult to learn? Why? Experience?

*General (RQ3)*

1.  Where do you see the greatest opportunities in teaching bilingual students? Why?
2.  Where do you see the biggest problems with teaching bilingual students? Why?

## Note

[1] Following Dewaele (2018), and in recognition of the fact that many English learners in Austria are learning the language not as a second but as a third or fourth language, we use the term 'LX English learner' to refer to these multilingual students. Similarly, students from a migration background in Austria are often learning German not as a second but as a third or sometimes fourth language, and we therefore refer to them as 'LX German speakers'.

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
