# Peer review of "Teaching English to Linguistically Diverse Students from Migration Backgrounds: From Deficit Perspectives to Pockets of Possibility"

_languages, doi:10.3390/languages7030186_

Round 1
Reviewer 1 Report
The manuscript is meant to analyze how some teachers that teach English in different types of Austrian schools perceive the multilingualism of their students in the English classes. Interviews of the teachers are used as the basis to investigate the main hypothesis of the study, i.e., teachers hold negative beliefs about their students’ multilingual backgrounds as they are not appropriate for English learning.
Overall, the manuscript is written correctly, and the theoretical approach adopted (with an extensive bibliography) seems to be appropriate to test the hypothesis. However, I have some concerns regarding the connection between the methodology and the analysis provided. Consequently, this and other aspects of the manuscript such as the following would require a revision:
Title
- lines 2-3: “highly diverse” should refer to “migration backgrounds” rather than to “multilingual students”.
Abstract
- line 12: “school” should be added to “secondary”: “secondary school teachers”.
Section 2. Literature review
- line 74: under the Literature review section, part of the previous research should include references where a more a positive viewpoint on multilingualism in the English classrooms as a pedagogical tool should be mentioned (e.g., Cummins, J. 2009. Multilingualism in the English-language classroom: Pedagogical considerations. TESOL Quarterly2: 317-321; Cenoz, J. & D. Gorter. 2011. A holistic approach to multicultural education: Introduction. The modern language journal 95.3: 339-343). This is only partly included in section 2.3 where translanguaging is shown as the stance supporting students’ multilingualism but in the references above not only translanguaging is the base of a supportive learning context in multilingual students.
- line 212: it should be clarified what “22-35 points” implies.
Section 3. Methods
- line 244-245: in RQ2 it should be clarified if “students” refers to only those with a linguistically diverse background or to all the students in the classroom.
- line 278: it would be convenient to clarify the following issues regarding the teacher participants: i) if all of them have a multilingual background as well; and ii) why only 6 participants were interviewed? It seems a too low number of participants to generalize the results and draw definite conclusions.
- line 298: under section 3.Interviews, it would be convenient to clarify the following issues: i) the relation between each set/themes of the questions included in the interviews and each Research Question and hypothesis; and ii) the existence of previous research on interviews about the attitudes of English teachers to multilingualism in education.
- line 331-332: it should be clarified if the typologies mentioned refer to the different subsections under the results section (e.g., 4.1.1-4.1.4).
- line 334: it should be clarified which additional categories emerged and why they differed from the themes intended to be explored through the interview questions.
Section 4. Results
General comment: it would be convenient to establish an explicit relation between the sections, the interview excerpts and/or the themes of the interview questions with the Research Questions posed in section 3.
- line 351: it should be clarified by the “statements about teachers’ satisfaction” are explored only in the Middle school section but not in the other two types of schools (i.e. sub-section 1.4. Teacher satisfaction only appears when analyzing the results from the Middle School teachers’ perceptions).
- line 357: the total number of students in each classroom should be mentioned (and this should be applied to the number of students from the other schools as well).
- line 361: the African country should be named (like the rest of the countries).
- line 363: it should be clarified here (and henceforth) if the students mentioned refer to all the students in the classroom or only those who have a multilingual background
- line 378: it should be clarified if LX is referring to L2 or L3.
- lines 455-456: it would have been convenient to develop the statement “those students should be able to take advantage of this [speaking multiple languages]”: did the teachers interviewed provide examples implemented in their own classroom practices? In fact, this is part of RQ3 (line 246) and it should have been taken this information into consideration to offer a more profound analysis of the topic. Only one explicit practice (focused on translaguaging) is mentioned later and described by Kerstin (lines 573-586).
- lines 465-466: it should be clarified in which schools the German language is used when teaching English (only English is spoken in English classes in the case of AHS? (lines 569-570)).
- lines 552: the idea of how students gain a lot from each other should be developed according to the results obtained from the interviews.
Section 5. Discussion
- lines 602-603: to assert that “At the VHS, the class atmosphere is reported to be equally friendly and open” seems to be a vague generalization of the results and a contradiction as on lines 478-479 it is affirmed that “Perceptions of students’ performance is also considerably more negative among these [VHS’s] teachers” and on line 500 “some negative views about students’ behaviour” is mentioned. Therefore, in the discussion section it would be convenient to make a distinction/contrast between students’ performance/behaviour and students’ group dynamics. Additionally, it should be made clear if both issues (i.e., performance/behaviour and group dynamics) relate to RQ3 or RQ2, or both [see my general comment].
- lines 712-717: the sentence is too long, and it should also be rephrased as the reasoning is difficult to follow.
Section 6. Conclusion
- line 802: only one specific “pedagogical practice” was mentioned and described by only one of the teachers. It would have been appropriate to have included a question about multilingual practices developed by the interviewed teachers as part of the “additional categories emerged” mentioned on line 334 and of the answers to RQ3.
Appendix A
- lines 953-955: it should be made clearer (see my comment on line 278.1) if all the teachers have multilingual backgrounds themselves and if this circumstance has an influence on their answers.
- line 966: the answers to question 2 were not reflected in the study, which would have been desirable as this contrast could have provided an interesting point of discussion about the role of migration in students’ performance/behaviour and students’ group dynamics in English classes.
- lines 977-978: the answers to questions 2 and 3 are not reflected in the study (see my comment on line 966).
Finally, very few minor issues related to grammatical expression should be addressed as well:
- line 292: “history” should be capitalized to keep consistency.
- line 296: in table 1 “music” and the last “female” should be capitalized to keep consistency.
- line 569: the article “the” should be added or pluralize “class” in “is not an issue in English classes…”.
- line 786: Teachers’
- lines 796-798: “Schools such as the MS featured in this study, where multilingualism and diversity are taken as the norm and students are performing above the national average, [comma should be inserted here] might be celebrated…”. It is a too long sentence though.
- line 972: perfoemance -- performance
Author Response
The manuscript is meant to analyze how some teachers that teach English in different types of Austrian schools perceive the multilingualism of their students in the English classes. Interviews of the teachers are used as the basis to investigate the main hypothesis of the study, i.e., teachers hold negative beliefs about their students’ multilingual backgrounds as they are not appropriate for English learning.
Overall, the manuscript is written correctly, and the theoretical approach adopted (with an extensive bibliography) seems to be appropriate to test the hypothesis. However, I have some concerns regarding the connection between the methodology and the analysis provided. Consequently, this and other aspects of the manuscript such as the following would require a revision:
Thanks so much for the constructive feedback. We hope that we could fully address all the comments and that the paper is now ready for publication.
Title
• lines 2-3: “highly diverse” should refer to “migration backgrounds” rather than to “multilingual students”.
We have changed the title to “Teaching English to linguistically diverse students from migration backgrounds” and now refer to linguistically diverse throughout.
Abstract
• line 12: “school” should be added to “secondary”: “secondary school teachers”.
We have added “secondary” here and are now using “secondary school” throughout.
Section 2. Literature review
• line 74: under the Literature review section, part of the previous research should include references where a more a positive viewpoint on multilingualism in the English classrooms as a pedagogical tool should be mentioned (e.g., Cummins, J. 2009. Multilingualism in the English-language classroom: Pedagogical considerations. TESOL Quarterly2: 317-321; Cenoz, J. & D. Gorter. 2011. A holistic approach to multicultural education: Introduction. The modern language journal 95.3: 339-343). This is only partly included in section 2.3 where translanguaging is shown as the stance supporting students’ multilingualism but in the references above not only translanguaging is the base of a supportive learning context in multilingual students.
We have included a short section on multilingualism research at the beginning of the literature review (including authors like Cummins, Duarte and Cenoz & Gorter), as suggested. We have also added a definition of translanguaging in Section 2.3.
• line 212: it should be clarified what “22-35 points” implies.
We have now provided the range of points that students could score on the test, and we hope that this clarifies what 22-35 points implies.
Section 3. Methods
• line 244-245: in RQ2 it should be clarified if “students” refers to only those with a linguistically diverse background or to all the students in the classroom.
We have added “multilingual” to RQ2 to clarify.
• line 278: it would be convenient to clarify the following issues regarding the teacher participants: i) if all of them have a multilingual background as well; and ii) why only 6 participants were interviewed? It seems a too low number of participants to generalize the results and draw definite conclusions.
None of the teachers had a multilingual background themselves and we have added the following to clarify this: “All interviewed teachers grew up monolingually with German and learned their other languages in instructional contexts.” We have also added the following to clarify why only six participants were interviewed and what this means for the generalizability of our results: “Of the 6-10 total English teachers at each school, six volunteered to be interviewed: two from the MS, three from the VHS and one from the AHS (see Table 1). While these participants cannot be considered to be representative of English teachers in these school types, their perspectives provide a glimpse into the experiences of teaching English to diverse multilingual students in this region.” Schools don’t have that many English teachers. We think that this is a reasonable number for a small-scale qualitative study. However, we have hedged our findings more and added some limitations to the Methodology and Discussion.
• line 298: under section 3.Interviews, it would be convenient to clarify the following issues: i) the relation between each set/themes of the questions included in the interviews and each Research Question and hypothesis; and ii) the existence of previous research on interviews about the attitudes of English teachers to multilingualism in education.
We have now added information as to how the sets of questions relate to the research questions. We have also included a reference to another interview study about attitudes of English teachers to multilingualism in education (Cephe & Yalcin 2015).
• line 331-332: it should be clarified if the typologies mentioned refer to the different subsections under the results section (e.g., 4.1.1-4.1.4).
We have made substantial changes to the organization of the results section, and the different subsections under the results section now refer to our three research questions. We have clarified this in the paper.
• line 334: it should be clarified which additional categories emerged and why they differed from the themes intended to be explored through the interview questions.
We have added the following information to clarify what kinds of additional categories emerged: “For example, although the intended focus of the interviews was language, teachers also mentioned topics like students’ behavior, the school environment, their own job satisfaction etc.” We hope that this is sufficiently clear now.
Section 4. Results
General comment: it would be convenient to establish an explicit relation between the sections, the interview excerpts and/or the themes of the interview questions with the Research Questions posed in section 3.
We have now made the relation between the sections, the themes of the interview questions and the research questions explicit throughout the document. As mentioned above, we have also revised the results section to align with the research questions.
• line 351: it should be clarified by the “statements about teachers’ satisfaction” are explored only in the Middle school section but not in the other two types of schools (i.e. sub-section 1.4. Teacher satisfaction only appears when analyzing the results from the Middle School teachers’ perceptions).
Teacher satisfaction is one of the additional themes that emerged despite not having a specific question to prove this information. To clarify, we have therefore mentioned that job satisfaction was one of these additional topics. Since this additional topic only came up in the middle school teachers’ perceptions, we only report it for middle school teachers. We have also reorganized the results sections, so that there is no longer a separate sub-section on teacher satisfaction. Instead, these additional insights are now presented in the sub-section on “Students’ abilities and the learning environment”. We hope that this clarifies the issue.
• line 357: the total number of students in each classroom should be mentioned (and this should be applied to the number of students from the other schools as well).
On average there are usually around 25 students in classrooms in Austria, but we do not know the number of students in these teachers’ many different classrooms. We think that this information is not relevant here and so we have not included it.
• line 361: the African country should be named (like the rest of the countries).
The exact quote from the transcript is “There are two from Africa. Actually, really only one.” Since the African country is not named and since this refers to only one or two students, we have deleted the sentence so that we can consistently name countries rather than regions or continents.
• line 363: it should be clarified here (and henceforth) if the students mentioned refer to all the students in the classroom or only those who have a multilingual background
We have mentioned in the contextual description what percentage of students on average come from a migration background in the schools. The statements in reference to students refer to all of them in the school environment.
• line 378: it should be clarified if LX is referring to L2 or L3.
Depending on the individual student, LX can refer to L2, L3 or L4 here. To clarify, we have added the following footnote the first time we mention LX in the paper: “Following Dewaele (2017), and in recognition of the fact that many English learners in Austria are learning the language not as a second but as a third or fourth language, we use the term ‘LX English learner’. Similarly, students from a migration background in Austria are often learning German not as a second but as a third or sometimes fourth language, and we therefore refer to them as ‘LX German speakers/students’.”
• lines 455-456: it would have been convenient to develop the statement “those students should be able to take advantage of this [speaking multiple languages]”:
We changed the text to the following: “Despite this, all three VHS teachers saw a clear advantage of their students’ multilingual abilities and felt that it was something that should be emphasized more. Beate believes that as populations become increasingly diverse, so does the demand for people who speak multiple languages and that the many linguistically diverse students at their school should be able to take advantage of this (pos. 251). Alena adds that these qualities are not adequately conveyed to multilingual students in schools or the administrative bodies, where their linguistic repertoire could be of value (pos. 252).”
• did the teachers interviewed provide examples implemented in their own classroom practices? In fact, this is part of RQ3 (line 246) and it should have been taken this information into consideration to offer a more profound analysis of the topic. Only one explicit practice (focused on translaguaging) is mentioned later and described by Kerstin (lines 573-586).
We have changed the text to make clearer that the teachers did not regularly engage in multilingual practice. As mentioned elsewhere, RQ 3 has also been rephrased and the findings in relation to it have been restructured.
• lines 465-466: it should be clarified in which schools the German language is used when teaching English (only English is spoken in English classes in the case of AHS? (lines 569-570)).
For the most part, teachers did not mention to what extent German may be used while teaching English. The AHS teacher does mention specifically that only English is used in English classes and the VHS teachers seem to rely on German translations to teach English vocabulary, but there is no mention of the use of German in the English classroom in the NMS interviews. To clarify the situation for VHS, we have made the following changes to the current paragraph: “This has implications for the English classroom as teachers seem to rely on students’ German knowledge, for example, for vocabulary acquisition. For example, Alena (pos. 230) mentions that particularly at the higher levels, it is difficult to teach subtle differences in vocabulary because students do not understand the subtle differences in the German translations of the vocabulary items either.”
• lines 552: the idea of how students gain a lot from each other should be developed according to the results obtained from the interviews.
We have added the following information from the interview that elaborates how students gain from each other: “She explains that students in these classes gain a lot from each other in terms of cultural understanding and that they are willing, and even interested, to learn what their class-mates’ native languages are like.”
Section 5. Discussion
• lines 602-603: to assert that “At the VHS, the class atmosphere is reported to be equally friendly and open” seems to be a vague generalization of the results and a contradiction as on lines 478-479 it is affirmed that “Perceptions of students’ performance is also considerably more negative among these [VHS’s] teachers” and on line 500 “some negative views about students’ behaviour” is mentioned. Therefore, in the discussion section it would be convenient to make a distinction/contrast between students’ performance/behaviour and students’ group dynamics. Additionally, it should be made clear if both issues (i.e., performance/behaviour and group dynamics) relate to RQ3 or RQ2, or both [see my general comment].
We have revised this section to discuss students’ learning and then the classroom environment. The statements about the classroom atmosphere are now more in line with the data.
lines 712-717: the sentence is too long, and it should also be rephrased as the reasoning is difficult to follow.
We have split up the sentence into several shorter sentences and added additional information so that our reasoning is easier to follow.
Section 6. Conclusion
• line 802: only one specific “pedagogical practice” was mentioned and described by only one of the teachers. It would have been appropriate to have included a question about multilingual practices developed by the interviewed teachers as part of the “additional categories emerged” mentioned on line 334 and of the answers to RQ3.
This point is covered under RQ3: What are their reported classroom practices and perspectives regarding linguistic diversity? Multilingual practice was thus one of the themes and not an additional category that emerged.
Appendix A
• lines 953-955: it should be made clearer (see my comment on line 278.1) if all the teachers have multilingual backgrounds themselves and if this circumstance has an influence on their answers.
As mentioned above, we now mention that none of the teachers interviewed come from multilingual backgrounds.
• line 966: the answers to question 2 were not reflected in the study, which would have been desirable as this contrast could have provided an interesting point of discussion about the role of migration in students’ performance/behaviour and students’ group dynamics in English classes.
We have not included information from this question as none of the answers spoke to the role of migration in students’ performance/behavior and students’ group dynamics in English classes. Answers to question 2 focused on how preparation-intensive teachers’ other subjects were and how much time teachers spent on grading assignments in English vs. their other subject. This data was not relevant to the current study.
• lines 977-978: the answers to questions 2 and 3 are not reflected in the study (see my comment on line 966).
The answers to the first part of question 3 are given in Table 1. Answers for the second part of question 3 (whether teachers had previously taught at another school) have been omitted as they were not relevant for the current study.
Finally, very few minor issues related to grammatical expression should be addressed as well:
• line 292: “history” should be capitalized to keep consistency.
We have removed all mentions of teachers’ other subjects (including History) to ensure that teachers would remain anonymous.
• line 296: in table 1 “music” and the last “female” should be capitalized to keep consistency.
The word “music” has been removed (since we have removed all mentions of teachers’ other subjects). For consistency, we now use no capitalization for the words “male” and “female”.
• line 569: the article “the” should be added or pluralize “class” in “is not an issue in English classes…”.
Thanks, we have changed this to “English classes”.
• line 786: Teachers’
Thanks, we have fixed that.
• lines 796-798: “Schools such as the MS featured in this study, where multilingualism and diversity are taken as the norm and students are performing above the national average, [comma should be inserted here] might be celebrated…”. It is a too long sentence though.
We have inserted the comma as requested and revised the sentence.
• line 972: perfoemance – performance
Thanks, we have corrected this.
Reviewer 2 Report
Thank you for the opportunity to review this paper. As a teacher-educator in a small, working-class town with increasing numbers of primary and secondary students from migration backgrounds, I read this scholarship with a great deal of interest.
The article centers on the perspectives of English teachers in Austria surrounding the multilingualism and language learning of students with migration backgrounds. Via a typological analysis of interviews with six English teachers, the authors find that, although some teachers hold some deficit discourses, others demonstrate “pockets of possibility” surrounding multilingual practice. The authors argue such possibilities can and should be capitalized upon to foster students’ linguistic and cultural development, including “improving outcomes in English.”
A significant contribution of the paper is that the research is undertaken in a language learning context that is understudied in multiple ways (i.e., Austria; linguistically-diverse small town schools), and in relationship to an understudied population (e.g., students from migration backgrounds). While deficit perspectives surrounding students classified as English learners (and their multilingual practices) are well-documented amongst teachers in L2 English-dominant contexts, less is known about such perspectives in L3 (or LX) English learning and teaching contexts.
Additionally, the references are appropriate to the topic, and the literature review is complete (although see comment regarding one lit review recommendation).
The comments below are provided in the spirit of supporting the authors’ revisions, with an eye toward eventual publication, whether in this outlet or another. The editors and authors will find that most comments are quite simple, as they center on clarifying terms and concepts for readers.
- The literature review would benefit from a close read -- and revisions -- for redundancy/repetition of claims/topics.
- Subsection 2.3 discusses “pockets of possibility.” As this is a central concept in the paper (even appearing in the title), I would recommend a more explicit definition for readers, as is provided in the section on “deficit discourses.” Additionally, where does the term “pockets of possibility” originate? If the authors coined this term, I’d suggest stating so explicitly, while continuing to credit García and colleagues (as we’d expect for a topic such as this one.)
- Similarly, in subsection 2.3, the term “transformation” is briefly mentioned — and then appears several times later in the paper as a primary goal of teachers’ multilingual/multicultural practice. As such, this term should be clarified: what does “transformation” mean in this context? Transforming from what to what? I encourage authors to assume readers are unfamiliar with the nuances of translanguaging pedagogies (although most will be familiar with them on the surface.)
- Relatedly, in Lines 71-72, the authors write “thereby pointing to the ‘transformative potential’ of English language learning for these students and enhancing their learning outcomes.” (Outcomes in English learning are also mentioned at the end of the Abstract, and the end of the Conclusion.). However, data is not presented that makes a direct link between practices taken up by these teachers (their “pockets of possibility”) and their students’ English learning outcomes. After multiple re-reads, I’m not certain that “English learning outcomes” is part of the story that this paper is telling, anyway. It is enough, I’d argue, that transformative practice(s) foster multilingual and multicultural development — particularly since those are two primary areas of focus (more than English learning per se) throughout the Findings, Discussion, and Conclusion.
However, if the authors feel that the English learning outcomes is an important piece of the puzzle of their paper, I’d recommend they 1. Present data related to what teachers believe about English outcomes and 2. Point back to Section 2.4 (Multilingualism and ELT) at multiple points throughout the Discussion and Conclusion. (If #1 is not possible, then the arguments related to #2 should be more nuanced and robust so that connections are blatant for readers.)
- The authors mention “discourses of possibility” as well as “pockets of possibility.” How are these similar to/different from one another? Relatedly, it would support readers in their interpretations of the paper if authors signaled for readers (e.g., a sentence or a citation) how they understand the term “discourse” (e.g., I think the authors may align with Gee’s (2004) D/discourses, but I’m not certain), as this is key throughout the paper.
- The authors describe the town as “non-elite, linguistically diverse small-town.” Please clarify for readers what is meant by “non-elite.”
- The authors state that the town has “higher than average percentages of residents with migration backgrounds.” Can authors provide percentages, as well as averages, since readers may not necessarily have a sense of what is average/above average in Austrian contexts?
- The authors write “The MS can be categorized as a ‘Brennpunktschule’ -- a school in a social hotspot.” I am wondering about this choice of terms. “School in a social hot spot” rang odd to me. “Social hot spot” is commonly associated with supply chain management issues in English (e.g., https://scholar.google.com/scholar?hl=en&as_sdt=0%2C36&q=social+hotspot&btnG=&oq=social+) and, although the direct translation of Brennpunktschule is “hot spot school,” a mini-corpus-type search for how professional translators translate this term turns up items such as, “at-risk school,” “high-risk school,” and “schools in disadvantaged areas” (e.g., scroll to bottom of: https://www.linguee.de/deutsch-englisch/uebersetzung/brennpunktschule.html) — all of which ring much more common to my ear (although I disprefer the first two, due to their deficit orientation). I’d recommend discussing this more amongst the author team and with editors.
- The authors mention that some teacher-participants undertook internships, with the implication that their years of service in certain schools were impacted. Please clarify for readers who are unfamiliar with Austrian/European teacher education systems what the relationship is.
- In the Methods section, the authors write: “The researcher was originally from the area in which the data were collected and having personal contacts amongst the students and teachers -- took great precautions to ensure his participants’ well-being.” Since others were involved in data collection, and this is part of a larger study, it seems there are multiple researchers. Here, did the authors mean “the person who conducted interviews”? If not, then please provide other clarification for readers who would likely find it confusing to hear mention of only one researcher at this point.
- In the Findings and Discussion, the authors discuss teachers’ perspectives on student behavior/motivation. Is this intended to be in response to RQ2? If so, please make this more explicit for readers. If not, please make more explicit for readers how student behavior/motivation connects to the RQs more generally.
- Lines 631-632: Despite having any detailed knowledge of students’ proficiencies and repertoires, or linguistic knowledge of students’ heritage languages, teachers make judgments about their 632 knowledge of grammar of their heritage languages. ==> comprehension stop. I think this is “despite NOT having…”?
- Lines 636-637: Second, literacy in its most basic sense is typically acquired in school, not in the home, such that autochthonous students typically learn to 637 read and write German at school. ==> I recommend explicitly qualifying the beginning of this statement to Austria or the German language (i.e., literacy in its most basic sense is typically acquired in school in the Austrian context), because acquiring decoding and encoding skills (what these authors call “basic literacy”) happens in a wide variety of contexts worldwide, including outside of formal, school-based learning contexts.
- Lines 644-645: we identified in the interviews discourses of L3 English as a burden, especially at MS and VHS. => I didn’t see this in the data. Could the authors be more explicit about what data they interpret in this way?
- Lines 689-693: The first sentence states that teachers did not report using multilingual practice in English language teaching and did not use s’s linguistic repertoires. The next sentence begins by describing how one of the teachers brought s’s home languages into the classroom, which seems to contradict the first sentence. Could the authors please clarify?
- In the Conclusion, the authors list a variety of activities or practices that teachers did (or were trying) that the authors consider “pockets of possibility.” In several cases, the authors mention “teachers” when only one teacher took up a practice or activity. I would urge the authors to be clear about when an activity/practice was taken up by just 1 vs. 2+ teachers. This speaks to how common the activity/practice is VS. whether it should be considered a counter-case.
- As a teacher-educator, I interpreted this paper as encouraging me (and others in my role) in engaging future and current teachers in developing translanguaging pedagogical practices, etc. As such, one of the questions that I came away from this paper with was this: Other than teacher beliefs, what facilitates — and what shuts down — the creation of the kinds of pockets of possibility that are described here? This is likely beyond the scope of this paper, but I mention it as a point of dialoguing with this scholarship and a question potentially (?) worth exploration at some point? (For instance, I am aware that many of the practicing teachers I work with do encounter resistance from monolingual parents, from some of their own administrators, and sometimes from multilingual students themselves. I’d be interested to hear what [if anything] these researchers’ scholarship shows.)
- Lastly, a point to check with editors: whether readers of Languages will know what "LX" means, as it is used in this paper.
Author Response
Reviewer 2:
Thank you for the opportunity to review this paper. As a teacher-educator in a small, working-class town with increasing numbers of primary and secondary students from migration backgrounds, I read this scholarship with a great deal of interest.
The article centers on the perspectives of English teachers in Austria surrounding the multilingualism and language learning of students with migration backgrounds. Via a typological analysis of interviews with six English teachers, the authors find that, although some teachers hold some deficit discourses, others demonstrate “pockets of possibility” surrounding multilingual practice. The authors argue such possibilities can and should be capitalized upon to foster students’ linguistic and cultural development, including “improving outcomes in English.”
A significant contribution of the paper is that the research is undertaken in a language learning context that is understudied in multiple ways (i.e., Austria; linguistically-diverse small town schools), and in relationship to an understudied population (e.g., students from migration backgrounds). While deficit perspectives surrounding students classified as English learners (and their multilingual practices) are well-documented amongst teachers in L2 English-dominant contexts, less is known about such perspectives in L3 (or LX) English learning and teaching contexts.
Additionally, the references are appropriate to the topic, and the literature review is complete (although see comment regarding one lit review recommendation).
The comments below are provided in the spirit of supporting the authors’ revisions, with an eye toward eventual publication, whether in this outlet or another. The editors and authors will find that most comments are quite simple, as they center on clarifying terms and concepts for readers.
Thanks so much for your supportive and helpful review and for your encouragement and feedback! We will aim to emulate your clarity and spirit of support in our future reviews.
1. The literature review would benefit from a close read -- and revisions -- for redundancy/repetition of claims/topics.
We have carefully reviewed the literature review for redundancies/repetition.
2. Subsection 2.3 discusses “pockets of possibility.” As this is a central concept in the paper (even appearing in the title), I would recommend a more explicit definition for readers, as is provided in the section on “deficit discourses.” Additionally, where does the term “pockets of possibility” originate? If the authors coined this term, I’d suggest stating so explicitly, while continuing to credit García and colleagues (as we’d expect for a topic such as this one.)
We have explicitly defined “pockets of possibility” and made sure to indicate that we have coined the use of this term in this particular way.
3. Similarly, in subsection 2.3, the term “transformation” is briefly mentioned — and then appears several times later in the paper as a primary goal of teachers’ multilingual/multicultural practice. As such, this term should be clarified: what does “transformation” mean in this context? Transforming from what to what? I encourage authors to assume readers are unfamiliar with the nuances of translanguaging pedagogies (although most will be familiar with them on the surface.)
In our first use of the word “transformation”, we have specified from what to what. In other cases, we have chosen to avoid the term.
4. Relatedly, in Lines 71-72, the authors write “thereby pointing to the ‘transformative potential’ of English language learning for these students and enhancing their learning outcomes.” (Outcomes in English learning are also mentioned at the end of the Abstract, and the end of the Conclusion.). However, data is not presented that makes a direct link between practices taken up by these teachers (their “pockets of possibility”) and their students’ English learning outcomes. After multiple re-reads, I’m not certain that “English learning outcomes” is part of the story that this paper is telling, anyway. It is enough, I’d argue, that transformative practice(s) foster multilingual and multicultural development — particularly since those are two primary areas of focus (more than English learning per se) throughout the Findings, Discussion, and Conclusion.
Thank you for reminding us to make a stronger focus back to English. We hope that we have achieved this. We hope that we have strengthened the argument that embracing a translanguaging stance fosters confidence and the development of academic self concepts, and that this – in turn – will also have an effect on English language learning. But also that embracing multilingual pedagogies would have a further impact.
However, if the authors feel that the English learning outcomes is an important piece of the puzzle of their paper, I’d recommend they 1. Present data related to what teachers believe about English outcomes and 2. Point back to Section 2.4 (Multilingualism and ELT) at multiple points throughout the Discussion and Conclusion. (If #1 is not possible, then the arguments related to #2 should be more nuanced and robust so that connections are blatant for readers.)
In our related research, we have found a relationship between what might be termed a “translanguaging stance” and higher outcomes in English. but the data discussed here doesn’t show that specifically. We have made references to this research clearer.
7. The authors mention “discourses of possibility” as well as “pockets of possibility.” How are these similar to/different from one another? Relatedly, it would support readers in their interpretations of the paper if authors signaled for readers (e.g., a sentence or a citation) how they understand the term “discourse” (e.g., I think the authors may align with Gee’s (2004) D/discourses, but I’m not certain), as this is key throughout the paper.
We have edited the paper to consistently use “deficit perspectives”, “pockets of possibility”, and “possibility perspective”, which are now defined in the literature review. We have therefore edited out the uses of the term “discourse” and thus not provided a definition of this term. The remaining uses of “discourse” in the paper refer to media and political discourses, and we hope that our uses of “discourse” in these contexts are clear.
8. The authors describe the town as “non-elite, linguistically diverse small-town.” Please clarify for readers what is meant by “non-elite.”
By non-elite, we meant vocationally oriented and non-selective. We have chosen to just refer to these schools as being non-selective now.
9. The authors state that the town has “higher than average percentages of residents with migration backgrounds.” Can authors provide percentages, as well as averages, since readers may not necessarily have a sense of what is average/above average in Austrian contexts?
No data is available for the town itself, but we have added percentages and averages for the municipality in which the town is located. Specifically, we have added the following information to the text: “The town has about 8,200 residents and is defined by a long history of migration and providing domicile for refugees. The municipality of about 17,500 residents, in which the town is located, has higher than average percentages of residents with migration back-grounds, with 30.6% of residents born outside Austria (compared to 20.1% for Austria as a whole and 16.2% for Upper Austria; Statistik Austria, 2020). The vast majority of the municipality’s residents with a migration background live in the town where our study was undertaken.”
10. The authors write “The MS can be categorized as a ‘Brennpunktschule’ -- a school in a social hotspot.” I am wondering about this choice of terms. “School in a social hot spot” rang odd to me. “Social hot spot” is commonly associated with supply chain management issues in English (e.g., https://scholar.google.com/scholar?hl=en&as_sdt=0%2C36&q=social+hotspot&btnG=&oq=social+) and, although the direct translation of Brennpunktschule is “hot spot school,” a mini-corpus-type search for how professional translators translate this term turns up items such as, “at-risk school,” “high-risk school,” and “schools in disadvantaged areas” (e.g., scroll to bottom of: https://www.linguee.de/deutsch-englisch/uebersetzung/brennpunktschule.html) — all of which ring much more common to my ear (although I disprefer the first two, due to their deficit orientation). I’d recommend discussing this more amongst the author team and with editors.
Thank you. We borrowed the translation from a source cited, but we like yours more and have used it instead.
11. The authors mention that some teacher-participants undertook internships, with the implication that their years of service in certain schools were impacted. Please clarify for readers who are unfamiliar with Austrian/European teacher education systems what the relationship is.
We have tried to make clear that the teacher also did her internship (a required part of her teacher education) at this school, before she was formally employed there.
12. In the Methods section, the authors write: “The researcher was originally from the area in which the data were collected and having personal contacts amongst the students and teachers -- took great precautions to ensure his participants’ well-being.” Since others were involved in data collection, and this is part of a larger study, it seems there are multiple researchers. Here, did the authors mean “the person who conducted interviews”? If not, then please provide other clarification for readers who would likely find it confusing to hear mention of only one researcher at this point.
We have clarified that the fourth author was the researcher who conducted the interviews.
13. In the Findings and Discussion, the authors discuss teachers’ perspectives on student behavior/motivation. Is this intended to be in response to RQ2? If so, please make this more explicit for readers. If not, please make more explicit for readers how student behavior/motivation connects to the RQs more generally.
We have made a clearer link between the three RQs and the three sections in the Results section. We have done this by completely reorganizing the Results section, such that each subsection now presents results for one of the RQs, and by clearly specifying which RQ each section covers.
14. Lines 631-632: Despite having any detailed knowledge of students’ proficiencies and repertoires, or linguistic knowledge of students’ heritage languages, teachers make judgments about their 632 knowledge of grammar of their heritage languages. ==> comprehension stop. I think this is “despite NOT having…”?
Thank you. This typo has been corrected.
15. Lines 636-637: Second, literacy in its most basic sense is typically acquired in school, not in the home, such that autochthonous students typically learn to 637 read and write German at school. ==> I recommend explicitly qualifying the beginning of this statement to Austria or the German language (i.e., literacy in its most basic sense is typically acquired in school in the Austrian context), because acquiring decoding and encoding skills (what these authors call “basic literacy”) happens in a wide variety of contexts worldwide, including outside of formal, school-based learning contexts.
Thank you for mentioning this important point. We have adopted your suggestion.
16. Lines 644-645: we identified in the interviews discourses of L3 English as a burden, especially at MS and VHS. => I didn’t see this in the data. Could the authors be more explicit about what data they interpret in this way?
Here we have reminded the reader of when the MS teacher noted that “there are a lot of problems with German, and then you add English to the mix”), i.e. signaling that English can be perceived as an additional burden. We have also expanded another example from the results section that exemplifies this: “For example, Alena (pos. 230) mentions that particularly at the higher levels, it is difficult to teach subtle differences in vocabulary because students do not understand the subtle differences in the German translations of the vocabulary items either.” We hope that this is clearer now.
17. Lines 689-693: The first sentence states that teachers did not report using multilingual practice in English language teaching and did not use s’s linguistic repertoires. The next sentence begins by describing how one of the teachers brought s’s home languages into the classroom, which seems to contradict the first sentence. Could the authors please clarify?
We have rephrased this and we hope the confusion here has been cleared up. To clarify, the teachers did not use any cross-linguistic multilingual pedagogies, but one of them did allow students to talk about cultural things from their heritage and teach expressions in their language.
18. In the Conclusion, the authors list a variety of activities or practices that teachers did (or were trying) that the authors consider “pockets of possibility.” In several cases, the authors mention “teachers” when only one teacher took up a practice or activity. I would urge the authors to be clear about when an activity/practice was taken up by just 1 vs. 2+ teachers. This speaks to how common the activity/practice is VS. whether it should be considered a counter-case.
Thank you. We have made more specific the practices only being reported by one of the six teachers.
19. As a teacher-educator, I interpreted this paper as encouraging me (and others in my role) in engaging future and current teachers in developing translanguaging pedagogical practices, etc. As such, one of the questions that I came away from this paper with was this: Other than teacher beliefs, what facilitates — and what shuts down — the creation of the kinds of pockets of possibility that are described here? This is likely beyond the scope of this paper, but I mention it as a point of dialoguing with this scholarship and a question potentially (?) worth exploration at some point? (For instance, I am aware that many of the practicing teachers I work with do encounter resistance from monolingual parents, from some of their own administrators, and sometimes from multilingual students themselves. I’d be interested to hear what [if anything] these researchers’ scholarship shows.)
Thank you for the suggestions and encouragement here. We have added a sentence about the focus of future research and a bit more insight from our own work about how to further support the development of possibilities.
20. Lastly, a point to check with editors: whether readers of Languages will know what "LX" means, as it is used in this paper.
We have included a footnote to define LX.
Reviewer 3 Report
There is no doubt that more research is needed on English teaching in multilingual classes and on the role of the students' L1, and as such this survey of teacher attitudes and views with six interviewees from three differing schools does provide interesting data which an make for an interesting article. However, I feel that the authors' in this article go beyond the limitations of the data with comments and conclusions that verge upon the normative, e.g. part of the title "From deficit discourses to pockets of possibility". Since there is no corroborative data such would conclusions/arguments would need to be better grounded in other research than is the case. In addition, the limitations of this study should also be made clear.
With regard to language I have notes som overuse of the progressive form, overly long sentences, in particular initially, and later the use of a number of unduly esoteric terms. The overall style is also at times non-idiomatic, and should be addressed.
In the methods, I am concerned whether the data in Table 1, the this to say the combination of school type and the teachers' linguistic repertoire would let readers with local knowledge deduce the respondents' identities. Further, should not the limitations of this study be addressed, either here and/or in the discussion?
When reading the Introduction I noted that the article seems unduly focused on Austria from the outset, while I think the article would have benefited from being situated in an international context before this local focus. I also missed an explicit research aim.
The review is comprehensive, but there seems to be a bit of overlap between sections. The many references to the authors' other articles, such as to provide additional information about Austria, should be avoided - an article should be able to stand alone. They should instead be part of the review. I would also contend that more work should be done to to build up to and frame the research questions, which I feel should come at the end of the review, not, confusingly for the reader, in the methods section.
The findings are interesting, and in my opinion are adequate for an interesting article focusing on teachers' views and attitudes towards the topic. However, the data is presented in an unduly descriptive manner, and splitting it between the three school types puts even further demands on the reader and thereby detracts from readability. I argue the need to present the findings an a more analytical manner, and perhaps using tables to bring out the differences between the three schools for instance.
The discussion and conclusion are long and detailed. However, as I have mentioned, I keep wondering whether the authors go beyond the fairly limited data on teacher views and attitudes in their arguments. For instance, the expression "pockets of possibility" keeps being repeated, and I think this needs to be better grounded in other studies than is the case, given the limitations of the focus of the data on the teachers' views and attitudes. In other words, the discussion would benefit from being more closely linked to the data.
A final point, the teaching of English does not seem to be particularly prominent in the article. If one looks at Research questions on page 5, lines 242 to 245, English is not even mentioned.
Author Response
Many thanks for this helpful review!
There is no doubt that more research is needed on English teaching in multilingual classes and on the role of the students' L1, and as such this survey of teacher attitudes and views with six interviewees from three differing schools does provide interesting data which an make for an interesting article. However, I feel that the authors' in this article go beyond the limitations of the data with comments and conclusions that verge upon the normative, e.g. part of the title "From deficit discourses to pockets of possibility". Since there is no corroborative data such would conclusions/arguments would need to be better grounded in other research than is the case. In addition, the limitations of this study should also be made clear.
The limitations of the study have been added to both the Methods section and the Discussion. Arguments about the pockets of possibilities that this particular context offered have been both strengthened and hedged.
With regard to language I have notes som overuse of the progressive form, overly long sentences, in particular initially, and later the use of a number of unduly esoteric terms. The overall style is also at times non-idiomatic, and should be addressed.
We did not find any particular “overuses” of the progressive, but we did thoroughly edit the article and cut down long sentences and the use of language which might be deemed “esoteric”.
In the methods, I am concerned whether the data in Table 1, the this to say the combination of school type and the teachers' linguistic repertoire would let readers with local knowledge deduce the respondents' identities. Further, should not the limitations of this study be addressed, either here and/or in the discussion?
With regard to identifying participants, we are confident that the area is big enough that even those with local knowledge will not be able to identify them. However, to err on the side of caution, we have left out information about participants’ second subjects. On reflection, this information does not add much to the study. We have also added limitations of the study both here and in the discussion section.
When reading the Introduction I noted that the article seems unduly focused on Austria from the outset, while I think the article would have benefited from being situated in an international context before this local focus. I also missed an explicit research aim.
The abstract and introduction have been edited to make the international relevance of this study. The research aim has been fore fronted.
The review is comprehensive, but there seems to be a bit of overlap between sections. The many references to the authors' other articles, such as to provide additional information about Austria, should be avoided - an article should be able to stand alone. They should instead be part of the review. I would also contend that more work should be done to to build up to and frame the research questions, which I feel should come at the end of the review, not, confusingly for the reader, in the methods section.
We moved the research questions to the introduction and have enhanced the framing of the literature review so that it more clearly identifies the need to answer these questions.
The findings are interesting, and in my opinion are adequate for an interesting article focusing on teachers' views and attitudes towards the topic. However, the data is presented in an unduly descriptive manner, and splitting it between the three school types puts even further demands on the reader and thereby detracts from readability. I argue the need to present the findings an a more analytical manner, and perhaps using tables to bring out the differences between the three schools for instance.
The organization of the findings has been changed so that there is one section per research question. We have not included tables, as we were not sure how to do so for qualitative data. Arguments have been made more closely linked to the data.
The discussion and conclusion are long and detailed. However, as I have mentioned, I keep wondering whether the authors go beyond the fairly limited data on teacher views and attitudes in their arguments. For instance, the expression "pockets of possibility" keeps being repeated, and I think this needs to be better grounded in other studies than is the case, given the limitations of the focus of the data on the teachers' views and attitudes. In other words, the discussion would benefit from being more closely linked to the data.
The discussion was edited down. The term “pockets of possibility” has been more robustly defined at the beginning and is used more sparingly in the discussion.
A final point, the teaching of English does not seem to be particularly prominent in the article. If one looks at Research questions on page 5, lines 242 to 245, English is not even mentioned.
The relevance of this exploration to English language teaching has been enhanced.
Round 2
Reviewer 1 Report
As most of the clarifications have been addressed, the manuscript has been significantly improved.
Author Response
As this reviewer has no further points of critique or suggestions for development, we have not changed anything in the article in response to this review.
Reviewer 3 Report
I am quite happy with the article now, it is far better balanced and far more readable. With regard to the latter, I would suggest that the authors in the results sections could improve this further by for instance using more explicit topic sentences to help the readers through the information dense subsections. I kept wishing for some subheadings here and there.
Next, considering exoteric language, most has been removed or changed, although I still noted the repeated use of the word "autochthonous". I suggest replacing this quite esoteric :-) word.
There were a few minor errors, such as words being split.
Author Response
Thanks again for these comments. We reviewed the Results section and found that each section and paragraph already started with a topic sentence. However, added sub-headings here, which we agree made the text much more readable. We kindly request that the editors check that these sub-sections align with the journal's guidelines.
We added a brief definition of the word 'autochthonous' when we first use it in the text, and have left the other uses. We feel like its use -- which is common in the German-speaking context, is preferable to other expressions like "native Austrian", etc.
We did not find any examples of words being inappropriately split, so we have not corrected that.